# A kinase-dependent checkpoint prevents escape of immature ribosomes into the translating pool

Melissa D. Parker[ID][1], Jason C. Collins[ID][1¤a], Boguslawa Korona[1¤b], Homa Ghalei[ID][1¤c], Katrin Karbstein[ID][1,2]*

**1** Department of Integrative Structural and Computational Biology, The Scripps Research Institute, Jupiter, Florida, United States of America, **2** HHMI Faculty Scholar, Chevy Chase, Maryland, United States of America

¤a Current address: Stem Cell Biochemistry Unit, National Institute of Dental and Craniofacial Research, Bethesda, Maryland, United States of America
¤b Current address: Division of Cellular and Molecular Pathology, Department of Pathology, University of Cambridge, Cambridge, United Kingdom
¤c Current address: Department of Biochemistry, Emory University School of Medicine, Atlanta, Georgia, United States of America
* kkarbst@scripps.edu

**Data Availability Statement:** All relevant data are within the paper and its Supporting Information files.

## Abstract

Premature release of nascent ribosomes into the translating pool must be prevented because these do not support viability and may be prone to mistakes. Here, we show that the kinase Rio1, the nuclease Nob1, and its binding partner Pno1 cooperate to establish a checkpoint that prevents the escape of immature ribosomes into polysomes. Nob1 blocks mRNA recruitment, and rRNA cleavage is required for its dissociation from nascent 40S subunits, thereby setting up a checkpoint for maturation. Rio1 releases Nob1 and Pno1 from pre-40S ribosomes to discharge nascent 40S into the translating pool. Weak-binding Nob1 and Pno1 mutants can bypass the requirement for Rio1, and Pno1 mutants rescue cell viability. In these strains, immature ribosomes escape into the translating pool, where they cause fidelity defects and perturb protein homeostasis. Thus, the Rio1–Nob1–Pno1 network establishes a checkpoint that safeguards against the release of immature ribosomes into the translating pool.

## Introduction

To maintain and balance protein levels within cells to support life, ribosomes must ensure that mRNA codons are faithfully translated into functional proteins. To guarantee their accurate function, the cell has to safeguard ribosome integrity during both assembly and its functional cycle. Ribosome assembly is a highly regulated process involving the proper folding and processing of 4 rRNAs, as well as the binding of 79 ribosomal proteins. Assembly is facilitated by over 200 transiently binding assembly factors that promote assembly and quality control and prevent immature ribosomes from initiating translation prematurely [1–4].

**Funding:** This work was supported by National Institute of Health/NIGMS (https://www.nigms.nih.gov) grant R01-GM086451 and Howard Hughes Medical Institute (https://www.hhmi.org) Faculty Scholar Grant 55108536 (to KK). The funders had no role in study design, data collection and analysis, decision to publish, or preparation of the manuscript.

**Competing interests:** The authors have declared that no competing interests exist.

**Abbreviations:** AMPPNP, adenylyl-imidodiphosphate; Gal, galactose; ITS1, internal transcribed spacer 1; MBP, maltose-binding protein; PIN, PilT-N-terminus; rDNA, ribosomal DNA; RENT, regulator of nucleolar silencing and telophase exit; snRNA, small nuclear RNA; TAP, tandem affinity protein; TCGA, The Cancer Genome Atlas; WT, wild-type.

To prevent misassembled ribosomes from reaching the translating pool, the precursor small (pre-40S) ribosomal subunit undergoes a series of quality-control checkpoints during late cytoplasmic maturation that verify proper ribosomal structure and function [5–7]. The importance of these checkpoints for cellular function is illustrated by the numerous diseases caused by haploinsufficiency or mutations in ribosomal proteins and assembly factors. These alterations dysregulate ribosome concentrations and/or lead to misassembled ribosomes and an increased propensity of patients to develop cancer [8–13].

One of the final steps in the biogenesis of 40S subunits in yeast is the maturation of the 3′-end of 18S rRNA from its precursor, 20S pre-rRNA. This step is carried out by the essential endonuclease Nob1 [14–17] and is promoted by its direct binding partner Pno1 [18]. Pno1 also blocks the premature incorporation of Rps26, as these two proteins occupy the same location on nascent or mature ribosomes, respectively [19–23].

Rio1 is an essential aspartate kinase bound to very late cytoplasmic pre-40S subunits that have shed all bound assembly factors except Nob1 and Pno1 [24–28]. Depletion of Rio1 or overexpression of a catalytically inactive Rio1 mutant leads to the accumulation of 20S pre-rRNA and assembly factors in 80S-like ribosomes [25, 28–30]. However, the role Rio1 plays in 18S rRNA maturation and ribosome assembly remains unknown, despite its interest as a target for the development of anticancer drugs [31–35] and the observation that mutations in the human homolog, RIOK1, accumulate in human cancers (The Cancer Genome Atlas [TCGA] Research Network: https://www.cancer.gov/tcga).

In this study, we use a combination of biochemical and genetic experiments to dissect the role of Rio1 in ribosome assembly. Our data show that Nob1 blocks the premature entry of nascent 40S subunits into the translating pool and requires rRNA maturation for its dissociation from nascent 40S subunits, thereby ensuring that only fully matured subunits engage in translation. Additionally, we provide evidence that Rio1 releases Nob1 and Pno1 from nascent ribosomes in an ATPase-dependent manner and that weak-binding Nob1 and Pno1 mutants can bypass the requirement for Rio1. Thus, the Rio1 kinase and Nob1 nuclease cooperate to restrict and regulate the entry of nascent ribosomes into the translating pool only after they are properly matured. Finally, bypassing Rio1 via self-releasing mutations in Pno1 or Nob1 results in release of immature ribosomes containing pre-rRNA into the translating pool. Together, these data reveal the function of a disease-associated kinase in licensing only the entry of mature ribosomes into the translating pool, thereby safeguarding the integrity of translating ribosomes.

## Results

### Nob1 inhibits mRNA recruitment

Nob1 is the endonuclease responsible for the final cleavage of pre-18S (20S) rRNA to produce its mature 3′-end [14–17]. Thus, in Nob1-depleted cells, ribosomes containing 20S pre-rRNA accumulate [14–17]. Surprisingly, however, in the Nob1-depleted cells, the 40S precursors enter the polysomes, which therefore do not collapse (Fig 1A) [5, 36], as is observed upon depletion of all other studied late 40S assembly factors [5, 19, 36]. This is surprising because Nob1 is an essential gene [37].

To test whether translation by immature ribosomes perturbs protein homeostasis, thereby affecting viability, we tested whether Nob1 depletion affected translational fidelity. These experiments take advantage of a collection of previously described luciferase reporter plasmids [38–41]. For these plasmids, firefly luciferase production depends on a mistranslation event. Although Nob1 depletion does not affect frameshifting, decoding, or stop codon recognition, start codon recognition is affected, leading to increased mistranslation at UUG codons relative

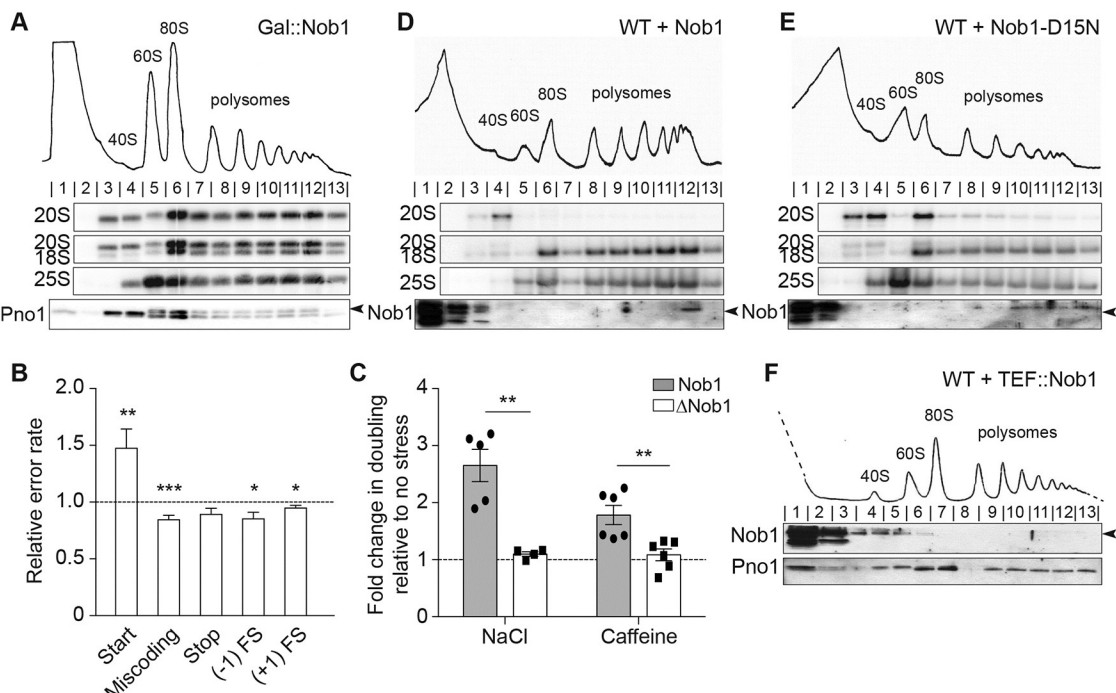

**Fig 1. Nob1 prevents entry of pre-40S subunits into the polysomes.** (A) Shown are 10%–50% sucrose gradients from lysates of cells depleted of Nob1 by growth in glucose for 12 h. Shown below the absorbance profile at 254 nm are northern blots of pre-18S rRNA (20S), mature 18S, and 25S rRNAs and western blots probing for Pno1. Arrowhead notes the upper band corresponding to Pno1. (B) The effects from depletion of Nob1 on the fidelity of start codon recognition, decoding, stop codon recognition, and FS (−1 and +1) were assayed using dual-luciferase reporters. Shown are the relative error rates of the glucose-depleted samples relative to replete samples. Data are the averages of 10–27 biological replicates, and error bars indicate the SEM. $^*p < 0.05$, $^{**}p < 0.01$, $^{***}p < 0.001$ by unpaired $t$ test. (C) Changes in doubling time in cells replete (Nob1) or depleted (ΔNob1) for Nob1 after exposure to high salt (1 M NaCl) or caffeine (10 mM). Values were compared to no-stress conditions (fold change = 1). Data are the averages of six (caffeine) or four to five (high salt) biological replicates, and error bars indicate SEM. $^{**}p < 0.01$ by unpaired $t$ test. Numerical data are listed in S1 Data. (D, E) Sucrose gradients of wild-type BY4741 cells overexpressing Nob1 (D) or Nob1-D15N (E) under the galactose-inducible, glucose-repressible Gal promoter in galactose for 12 h. Western blots probed for Nob1. Arrowhead notes the upper band corresponding to Nob1. (F) Sucrose gradients of WT BY4741 cells overexpressing Nob1 under the constitutive Tef2 promoter grown in glucose. See also S1 Fig. FS, frameshifting; Gal, galactose; WT, wild-type.

to AUG codons (Fig 1B). To test whether the remodeled proteome arising from these mis-translation events also affects stress resistance as previously observed from defects in translation arising from changes in ribosome composition or translation factors [7, 42, 43], we measured the effects from Nob1 depletion on growth in high-salt or caffeine-containing media. These data show that Nob1 depletion provides resistance to caffeine and high salt (Fig 1C), consistent with a perturbation in protein synthesis. Changes in stress resistance are not due to an activation of the general stress response, as read out by eIF2α phosphorylation (S1A Fig). Together, these data suggest that immature 40S ribosomes can perturb protein homeostasis, as observed for ribosomes lacking Rps26, Rps10, or Asc1 [7, 42, 44].

Given that upon Nob1 depletion, translation remains intact but viability is compromised due to changes in the outcomes of translation, we reasoned that mechanisms might exist that prevent the premature release of immature ribosomes. One possible way such a mechanism might work is if Nob1 itself blocks mRNA recruitment, and its dissociation requires its prior activity, as for many enzymes. To test such a mechanism, we used a dominant-negative, cata-lytically inactive mutant of Nob1 (Nob1-D15N) (S1B Fig). Nob1-D15N is a mutation in the conserved PilT-N-terminus (PIN) domain of Nob1, rendering Nob1 able to bind but not to

cleave its 20S pre-rRNA substrate [15, 17]. The accumulation of 20S pre-RNA in wild-type (WT) cells is noticeable after galactose (Gal)-promoter-driven overexpression of Nob1-D15N for 8 h but not in cells expressing an empty vector (S1C Fig).

To assess whether the Nob1-containing pre-40S ribosomes enter the polysomes, as observed for pre-ribosomes accumulated in the absence of Nob1, we performed polysome profiling followed by northern blotting on cells overexpressing Nob1 or Nob1-D15N. Overexpressing WT Nob1 results in 20S pre-rRNA concentrated only in the 40S fraction, whereas the polysomes contained only mature 18S rRNA (Fig 1D). In contrast to Nob1-depleted cells, very little 20S pre-rRNA escaped into the polysomes in Nob1-D15N-overexpressing cells and, instead, accumulated in pre-40S and 80S-like ribosome peaks (Fig 1E). This observation is also consistent with the appearance of robust polysomes in Nob1-depleted cells but not Nob1-D15N cells (Fig 1A and 1E). Thus, ribosomes containing immature 20S pre-rRNA can recruit mRNA to enter the polysomes in the absence of Nob1 but not in its presence, suggesting that Nob1 blocks mRNA recruitment.

If Nob1 blocks mRNA recruitment, then it should not be found in the polysomes. Consistently, Nob1 is not found in the polysomes of WT cells [5] or when expressed under the Tef2 promoter, which produces significantly more Nob1 than the endogenous promoter (Fig 1F).

We also considered the possibility that it is not the presence of Nob1 but, rather, its interacting partner Pno1 that blocks entry into the polysomes. However, we note that Pno1 can be found in the polysomes in Nob1-depleted cells, showing that Pno1 does not block polysome recruitment (Fig 1A). The finding that Pno1 remains bound to actively translating 20S-containing ribosomes in Nob1-depleted cells also explains why these translating ribosomes do not support growth. Pno1 blocks Rps26 binding [19, 20]. Thus, the remaining Pno1 will prevent binding of Rps26, an essential protein required for translation of ribosome components [42].

## rRNA cleavage facilitates Nob1 release

If Nob1 release from nascent 40S requires rRNA cleavage by Nob1, then Nob1 blocking mRNA recruitment to premature ribosomes would enable a quality-control mechanism to ensure that only ribosomes containing matured rRNA enter the polysomes. To test whether Nob1-dependent rRNA cleavage facilitates its dissociation from ribosomes, we used a previously described quantitative in vitro RNA binding assay [16]. This assay measures the binding of Nob1 to mimics of the 20S pre-rRNA substrate (H44-A2), the 18S rRNA ribosome product (H44-D), or the internal transcribed spacer 1 (ITS1) 3′-product (D-A2) via native gel shift. The data show that Nob1 binds the substrate mimic and the 3′-ITS1 product with similar affinities ($K_d$ = 0.93 and 0.96 μM, respectively). In contrast, Nob1 binds the 18S rRNA mimic somewhat more weakly ($K_d$ = 1.89 μM) (Fig 2). These differences, albeit small, suggest that Nob1 predominantly interacts with ITS1, consistent with previous structure probing data [16, 27]. Furthermore, the data suggest that after Nob1 cleaves 20S pre-RNAs, Nob1 will remain bound to its 3′-cleavage product and not to the matured 18S rRNA. Thus, Nob1 blocks premature ribosomes from binding mRNA until rRNA is cleaved, which facilitates Nob1 dissociation from nascent ribosomes, thereby setting up a mechanism to ensure that only ribosomes with fully matured rRNA enter the translating pool.

## Rio1 authorizes translation initiation of nascent 40S ribosomes

The data above suggest that rRNA maturation promotes Nob1 dissociation from pre-40S subunits. Nevertheless, because Nob1 is also bound to Pno1 [18, 22], Nob1 release from pre-40S also requires separation from Pno1. Thus, to test whether other late-acting 40S assembly factors play a direct role in Nob1 release, we carried out a limited screen for factors whose

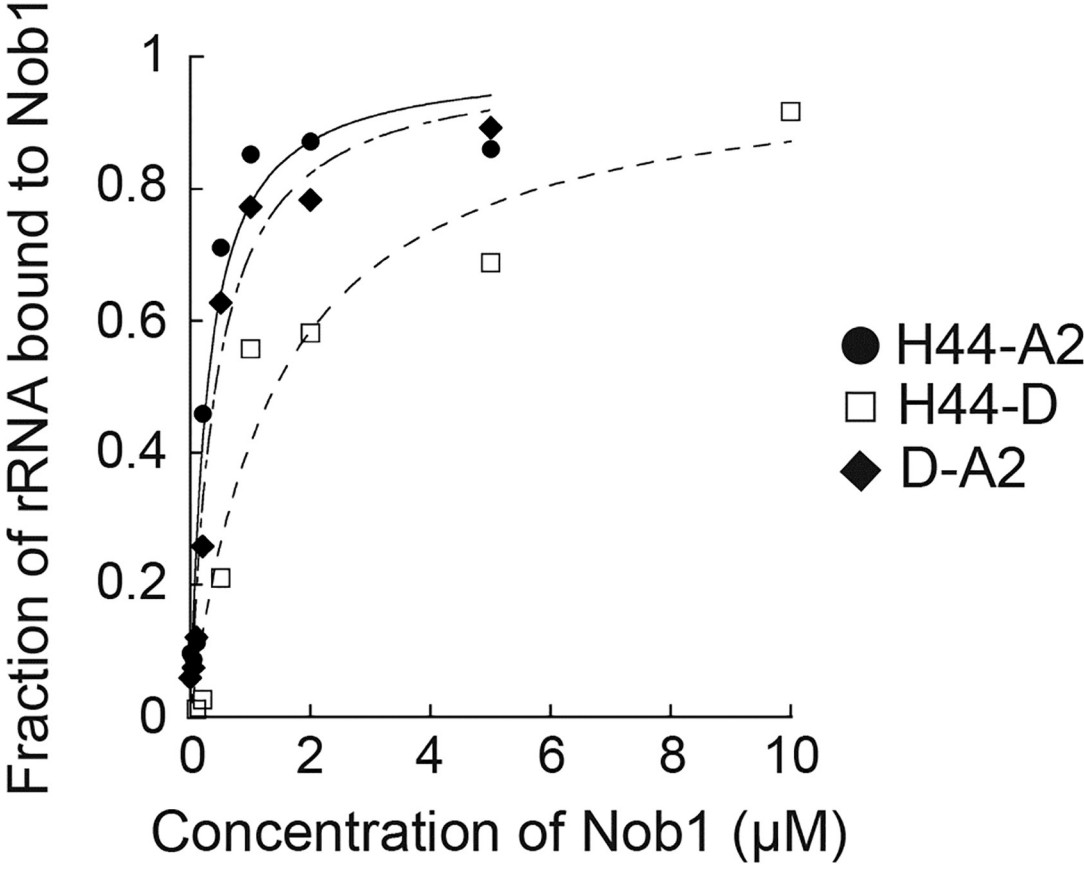

**Fig 2. Nob1 dissociates with the 3′-cleavage product following rRNA cleavage.** Representative RNA binding assay with in vitro transcribed H44-A2 (20S pre-rRNA mimic, black circles), H44-D (18S rRNA mimic, white squares), or D-A2 (3′-ITS1, black diamonds) RNAs and with recombinant Nob1. Four or five independent experiments yielded values of $K_d = 0.93 +/-$ 0.09 μM for Nob1 binding H44-A2, $K_d = 0.96 +/- 0.05$ μM for Nob1 binding D-A2, and $K_d = 1.89 +/- 1.04$ μM for Nob1 binding H44-D. Numerical data are listed in S1 Data. ITS1, Internal transcribed spacer 1.

overexpression rescues the dominant-negative phenotype from Nob1-D15N overexpression (S2 Fig). This screen showed that overexpression of the aspartate kinase Rio1 rescues the growth phenotype from Nob1-D15N overexpression (Fig 3A). Furthermore, Rio1 activity is required for this rescue because mutations that block phosphorylation, D261A (the phosphoaspartate) [25] and K86A (in the P-loop), did not rescue the Nob1-D15N growth phenotype (Fig 3A).

To test whether Rio1 overexpression promotes endonuclease activity of Nob1 and thus rescues the Nob1 mutation by "repairing" its catalytic activity, we carried out northern blot analysis. Overexpressing Nob1-D15N and Rio1 together resulted in a 6.5-fold increase in 20S pre-rRNA accumulation compared with Nob1-D15N alone (Fig 3B). This is the opposite of what would be expected if Rio1 enhances Nob1 activity. Additionally, overexpression of Rio1 did not rescue the lack of Nob1 (S3A Fig), as expected if Rio1's role is to release Nob1 rather than to promote rRNA cleavage. These data show that Rio1 does not rescue the growth phenotype of Nob1-D15N by stimulating rRNA cleavage.

To test whether, instead, Rio1 overexpression rescues the Nob1-D15N growth phenotype by releasing Nob1, thereby allowing 20S pre-rRNA-containing ribosomes to enter the translating pool as in Nob1-depleted cells, we used polysome profiling coupled with northern blot

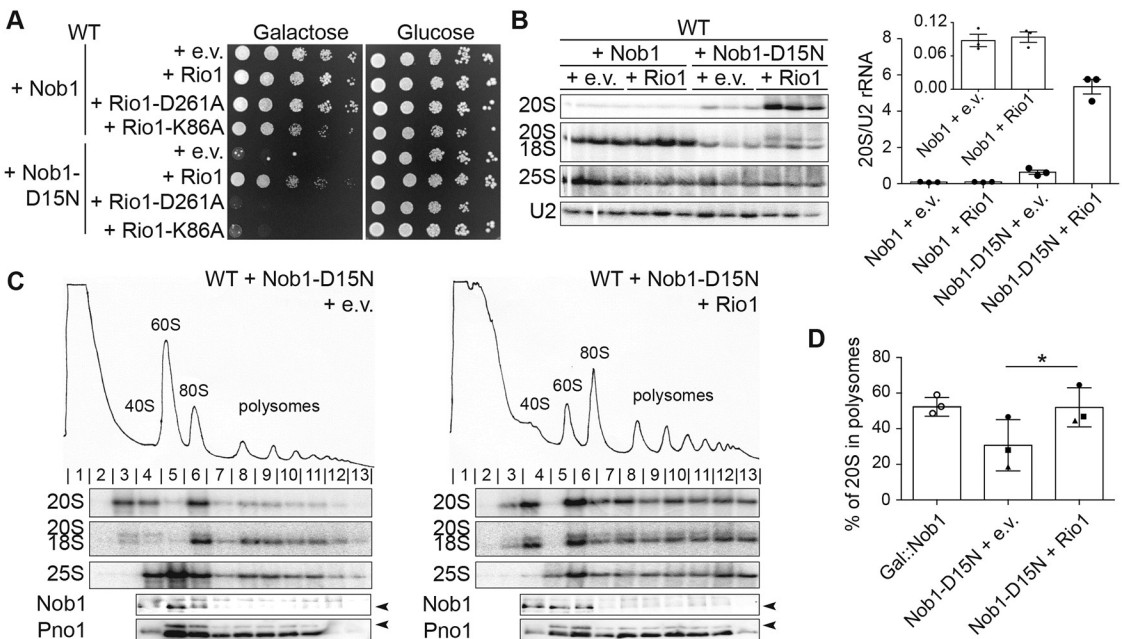

**Fig 3. Rio1 releases 40S ribosomes into the translating pool.** (A) Growth of cells containing an e.v. or WT Rio1, Rio1-D261A, or Rio1-K86A under a copper-inducible (Cup1) promoter and Nob1 or Nob1-D15N under the Gal promoter were compared by 10-fold serial dilutions on glucose or galactose dropout plates with 100 μM CuSO₄. (B) Left: northern blot analyses of total cellular RNA from cells in panel A grown in galactose with 100 μM CuSO₄ for 16 h. Right: quantification of the northern blot. The 20S pre-rRNA levels were normalized to U2 snRNA. (C) Shown are 10%–50% sucrose gradients from cell lysates of cells in panel B. Northern blots of 20S, 18S, and 25S rRNA and western blots probing for Nob1 and Pno1 are shown below the absorbance profile at 254 nm. Arrowheads note the bands corresponding to Nob1 (lower band) and Pno1 (upper band). (D) Quantification of the gradient northern blots in (C) and in Fig 1A. Percentage of 20S pre-rRNA in polysomes (fractions 8–13) compared with total 20S pre-rRNA was calculated. Data are the average of three biological replicates, and error bars indicate SEM. Samples grown and analyzed on the same day were considered paired replicates, as indicated by the circle, square, and triangle dots on the graph. Paired *t* test was used for statistical analysis; *p* = 0.0123. See also S1, S2 and S3 Figs. Numerical data are listed in S1 Data. e.v., empty vector; Gal, galactose; snRNA, small nuclear RNA; WT, wild-type.

analysis. As before, Nob1-D15N overexpression results in accumulation of 20S pre-rRNA in pre-40S and 80S-like ribosomes (Fig 3C, left), with only 30% of 20S pre-rRNA in polysome fractions (Fig 3D). Simultaneous Rio1 overexpression releases 20S pre-rRNA-containing ribosomes into the polysomes (Fig 3C, right), with a statistically significant increase to 52% of 20S pre-rRNA in polysomes (Fig 3D). The accumulation of pre-ribosomes in the translating pool when Rio1 is overexpressed in the Nob1-D15N background is the same as that observed upon Nob1 depletion (both 52%). These data show that Rio1 overexpression promotes the release of immature, 20S-containing ribosomes into the translating pool. Furthermore, the data suggest that this occurs via Nob1 release, thereby turning Nob1-D15N-containing ribosomes into Nob1-depleted ribosomes. This model is further supported by polysome analysis of Rio1-depleted cells, in which few 20S pre-rRNA-containing ribosomes reach the polysomes, instead accumulating in mRNA-free 80S-like assembly intermediates [5] (S1D Fig). Furthermore, the 20S-containing ribosomes that do enter the polysomes lack Nob1 (S1D Fig).

## Rio1 binds Nob1 and Pno1 directly and stimulates their release from pre-40S ribosomes

Rio1 is an atypical aspartate kinase. By analogy to its close relative Rio2, it is believed that the Rio1 functional cycle involves ATP binding, autophosphorylation, and subsequent

dephosphorylation, resulting in net ATP hydrolysis, which must be coupled to conformational changes in Rio1 or its binding partner(s) [45, 46]. Previous analyses suggest that Rio1 interacts with pre-40S ribosomes during the final cytoplasmic assembly steps when the pre-40S is bound only to Nob1 and its binding partner Pno1 [27, 28], consistent with our data that indicate a role for Rio1 in Nob1 release to allow for discharge of the nascent 40S subunits into the translating pool.

To test this model, we performed in vitro protein binding assays with recombinant Rio1, Nob1, and Pno1. These experiments show that maltose-binding protein (MBP)-Rio1 binds Nob1 and Pno1 but not either Nob1 or Pno1 individually, suggesting that Rio1 recognizes the Nob1–Pno1 complex (Fig 4A and S4A–S4C Fig). Importantly, the presence of adenylyl-imido-diphosphate (AMPPNP), a nonhydrolyzable ATP analog, is required for formation of the Rio1–Nob1–Pno1 complex because little to no complex formation is observed in the presence of ADP (Fig 4A and S4A–S4C Fig).

These data suggest that Rio1 recognizes the Nob1–Pno1 complex in an ATP-dependent manner. To test whether autophosphorylation (and therefore ATP hydrolysis) is responsible for breaking this complex, we developed an in vitro release assay using assembly intermediates purified from yeast and purified recombinant Rio1. In this assay, tandem affinity protein (TAP)-Pno1 ribosomes purified from cells depleted of Rio1 are incubated with Rio1 in the presence of ATP, the nonhydrolyzable ATP analog AMPPNP, or ADP. Release of assembly factors was monitored using an assay in which the reactions are layered onto a sucrose cushion to pellet ribosomes and all bound factors, whereas free proteins will be in the supernatant. Little Nob1 or Pno1 (8% and 5% of Nob1 or Pno1, respectively) were released in a mock incubation (Fig 4B and 4C). Addition of Rio1 alone, or in the presence of ADP or AMPPNP, increased this slightly (approximately 10% of Nob1 and 20% Pno1 released, respectively), whereas addition of Rio1 and ATP led to a 5–10-fold increase in the release of these assembly factors (35% Nob1 and 49% Pno1, respectively, Fig 4B and 4C). This finding demonstrates that Rio1 uses ATP hydrolysis to stimulate the dissociation of Nob1 and Pno1 from the pre-40S subunit. Nonetheless, addition of Nob1 and Pno1 to Rio1 does not affect the rate of ATP hydrolysis by Rio1 (S4D Fig). This suggests that catalytic activity by Rio1 has additional requirements, perhaps reading out rRNA cleavage. Additionally, or alternatively, ATP hydrolysis might be rate limited by hydrolysis of the phosphoaspartate, whereas release of Nob1 and Pno1 from pre-40S might only require Rio1 phosphorylation. Additional future experiments will be required to distinguish between these options.

## Weak-binding Nob1 and Pno1 mutants can bypass Rio1 activity

The data above show that Rio1 can release Nob1 and Pno1 from nascent 40S subunits in vitro. To confirm a role for Rio1 in the release of Nob1 and Pno1 from pre-40S ribosomes in vivo, we screened a collection of mutants in Pno1 and Nob1 for their ability to rescue the loss of cell viability upon Rio1 depletion. These included Pno1 mutants that disrupt the binding to Nob1 (GXXG, WK/A, HR/E, DDD/K) [18] (S5A Fig) or weakened its interactions with the ribosome (KKKF) [23], as well as mutations in Nob1 that weaken ribosome binding (truncations Nob1-434 and Nob1-363 and L88A/S89A, L93A/L96A, K320A/F322A, Q28R, D271N, D271R/F272A, Q280R, T225A, K80A, Y300A, and R303A) or Pno1 binding (W223G) [47] (S5C Fig). Finally, overexpression of Nob1 was also tested (S5D Fig). Of all of these mutants, only the weak-binding Pno1-KKKF (K208E/K211E/K213E/F214A) mutant was able to rescue the lethal phenotype from Rio1 depletion (Fig 4D, S5A and S5B Fig).

To confirm that Pno1-KKKF rescued Rio1 depletion, we analyzed pre-rRNA levels in cells expressing WT Pno1 or Pno1-KKKF in the presence or absence of Rio1 using northern

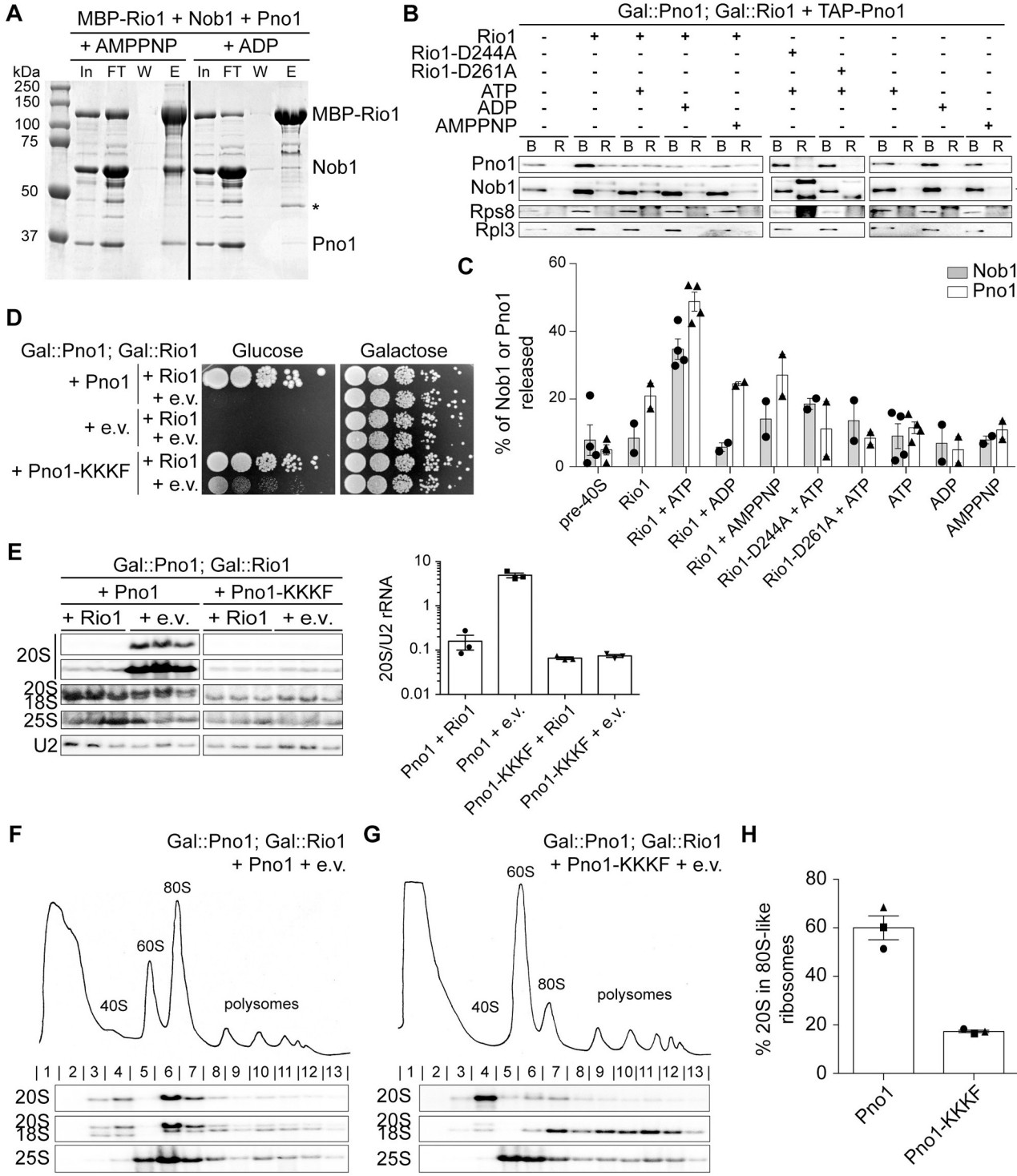

**Fig 4. Rio1 stimulates release of Nob1 and Pno1 from the pre-40S ribosome.** (A) Rio1 binds Nob1 and Pno1 in the presence of the nonhydrolyzable ATP analog AMPPNP. Shown are Coomassie-stained SDS-PAGE gels of protein binding assays using purified, recombinant MBP-Rio1, Nob1, and Pno1 in the presence of AMPPNP or ADP. The order of the samples was edited for clarity. *MBP. See also S4 Fig. (B) Western blot analysis of a release assay upon addition of purified, recombinant Rio1, Rio1-D244A, or Rio1-D261A and ATP, ADP, or AMPPNP. The arrowhead denotes the band corresponding to Nob1. (C) Quantification of the release assay western blot. The percent of Nob1 and Pno1 released from the ribosome compared with the total Nob1 or Pno1 in each sample. Data are the average of two to four biological replicates, and error bars indicate SEM. (D) Growth of cells expressing either WT Pno1 or Pno1-KKKF (K208E/K211E/K213E/F214A) and either an e.v. or Rio1 were compared by 10-fold serial dilutions on glucose or galactose dropout plates. See also S5 Fig. (E) Left: northern blot analyses of total cellular RNA from cells in panel D grown in glucose for 16 h. Samples were run on the same gel, and the order was edited for clarity. Right: 20S pre-rRNA accumulation was normalized to U2 snRNA in these cells.

(F, G) Shown are 10%–50% sucrose gradients of cell lysates of cells from panel E (Gal::Rio1; Gal::Pno1 cells expressing either WT Pno1 [F] or Pno1-KKKF [G] from plasmids). Shown below the absorbance profile at 254 nm are northern blots of 20S, 18S, and 25S rRNAs. (H) Quantification of the gradient northern blots in (F) and (G). Percentage of 20S pre-rRNA in 80S-like ribosomes (fractions 6–7) compared with total 20S pre-rRNA was calculated. Data are the averages of three biological replicates, as indicated by the circle, square, and triangle dots on the graph, and error bars indicate SEM. 80S-like assembly intermediates accumulate in Rio1-depleted cells containing WT Pno1 but not Pno1-KKKF. Numerical data are listed in S1 Data. AMPPNP, adenylyl-imidodiphosphate; B protein, ribosome-bound protein; E, elution; e.v., empty vector; FT, flow through; Gal, galactose; In, input; MBP, maltose-binding protein; R protein, released protein; snRNA, small nuclear RNA; TAP, tandem affinity protein; W, final wash; WT, wild-type.

blotting. These data showed a 30-fold increase in 20S pre-rRNA accumulation in cells lacking Rio1 compared with cells expressing WT Rio1 and WT Pno1. In contrast, with Pno1-KKKF, no 20S pre-rRNA accumulation was observed in the absence of Rio1 (Fig 4E). Finally, the accumulation of 80S-like assembly intermediates observed in the Rio1-depleted cells containing WT Pno1 (S1D Fig, Fig 4F and 4H) is rescued by the Pno1-KKKF mutation (Fig 4G and 4H). Together these data show that the function of Rio1 can be bypassed by self-release of Pno1 from the pre-40S ribosome.

### Bypass of Rio1 activity leads to release of immature 40S ribosomes into the translating pool

Although none of the weak-binding Nob1 mutants rescued the growth defects observed from Rio1 depletion, this could be explained if Pno1 remained bound to ribosomes after Nob1 had self-released, as it does in Gal::Nob1 cells (Fig 1A). This would block Rps26 recruitment [19, 20], leading to loss of viability. To test this idea, we analyzed polysome profiles from cells depleted of Rio1 and containing truncated Nob1-363 (in which the Nob1 gene does not encode for amino acids 364–459). Nob1-363 binds rRNA more weakly, as suggested by a growth phenotype from this mutant—which can be rescued when Nob1 is overexpressed from the TEF promoter (S6A Fig)—as well as RNA binding data (S6B Fig). Notably, the self-releasing Nob1-363 is not found in the 80S-like ribosomes that accumulate upon Rio1 depletion (Fig 5A), demonstrating that Rio1's role in Nob1 release can be bypassed by a weak-binding Nob1 mutant. In contrast, Pno1 is found in the polysomes in these cells (Fig 5A), further demonstrating the requirement for Rio1 in Pno1 release and explaining why bypass of Nob1 release does not rescue viability.

The data above provide strong evidence for a role for the kinase Rio1 in releasing Nob1 and Pno1 from nascent 40S subunits. Because Nob1 dissociation also requires prior Nob1-dependent rRNA cleavage, this pathway ensures only ribosomes containing fully matured rRNA are discharged into the translating pool. Thus, these data strongly support a role for Rio1 in ensuring only matured ribosomes enter the translating pool.

To directly test the importance of this control point in ensuring that only mature ribosomes enter the translating pool, we took advantage of the self-releasing Pno1-KKKF mutant, which bypasses the requirement for Rio1 and allows for cellular growth in the absence of Rio1 (Fig 4). If Rio1 restricts premature entry of immature ribosomes into the translating pool, we would predict that bypassing Rio1 with the Pno1-KKKF mutant would allow for the escape of immature ribosomes into the polysomes even if cells contain Rio1.

To test this prediction, we analyzed the rRNA in polysomes of cells expressing WT Pno1 or Pno1-KKKF. In cells expressing Pno1-KKKF, 15% of 20S pre-rRNA escaped into the polysomes compared with only 3% of 20S pre-rRNA in WT cells (Fig 5B–5D). Importantly, 20S pre-rRNA does not accumulate in the Pno1-KKKF mutant (Fig 4E). This finding confirms a role for the Pno1–Nob1 checkpoint in restricting the release of immature ribosomes into the translating pool.

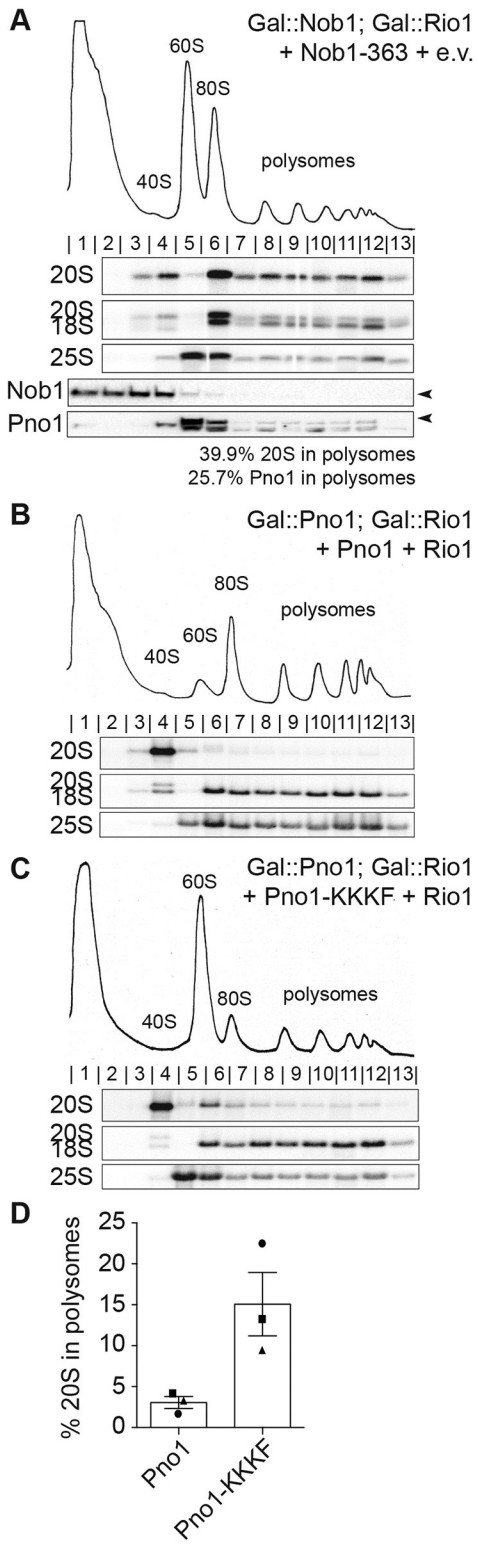

**Fig 5. Rio1-mediated quality control ensures only mature 40S are released into the translating pool.** (A) Sucrose gradients analyzed by northern and western blot of lysate from Gal::Nob1; Gal::Rio1 cells expressing the Nob1-363 truncated protein on a plasmid and grown in glucose for 16 h. Arrowheads note the bands corresponding to Nob1 and Pno1. The percentage of 20S pre-rRNA and Pno1 in polysomes (fractions 8–13) compared with total 20S pre-rRNA or Pno1 was calculated and listed below the western blot. Lane 5 was omitted from the Pno1 calculations because the

Pno1 band and the lower band were not separated enough in this lane to accurately quantify Pno1. (B, C) Shown are 10%–50% sucrose gradients of cell lysates from Gal::Rio1; Gal::Pno1 cells expressing WT Rio1 and either WT Pno1 (B) or Pno1-KKKF (C) from plasmids and grown in glucose for 16 h. Shown below the absorbance profile at 254 nm are northern blots of 20S, 18S, and 25S rRNAs. (D) Quantification of the gradient northern blots. The percentage of 20S pre-rRNA in polysomes (fractions 8–13) compared with total 20S pre-rRNA was calculated. Data are the average of three biological replicates, as indicated by the circle, square, and triangle dots on the graph, and error bars indicate SEM. See also S6 Fig. Numerical data are listed in S1 Data. e.v., empty vector; Gal, galactose; WT, wild-type.

## Discussion

Nascent 40S subunits arrive in the cytoplasm bound to seven assembly factors, which block premature translation initiation by immature assembly intermediates by preventing the association of translation initiation factors [19]. These assembly factors are then released in a series of regulated steps that form part of a translation-like cycle, which couples their release to quality-control steps [5–7]. Furthermore, when premature ribosomes do escape into the translating pool, they are unable to support cell viability [5, 36, 42]. Together, these observations demonstrate the importance of preventing premature translation initiation by immature ribosomes. The data herein demonstrate that the discharge of ribosomes into the translating pool is a regulated quality-control step during maturation of the small ribosomal subunit.

### Nob1 blocks mRNA recruitment to 20S pre-rRNA-containing ribosomes

Nob1 is the endonuclease that produces mature 18S rRNA [14–17]. Yeast lacking Nob1 or overexpressing a dominant-negative, inactive Nob1 (Nob1-D15N) both accumulate pre-18S rRNA (20S pre-rRNA). Nonetheless, 20S pre-rRNA-containing ribosomes enter the translating pool in cells lacking Nob1 [5, 36], whereas 20S pre-rRNA-containing ribosomes bound to Nob1-D15N do not bind mRNA. These data strongly suggest that the presence of Nob1 blocks recruitment of mRNAs to the nascent ribosome.

Nob1 cleaves 20S pre-rRNA endonucleolytically, yielding two products: the fully mature 40S subunit and the ITS1 product, which is subsequently degraded by the exonuclease Xrn1 [48]. Our data demonstrate that Nob1 binds more strongly to the ITS1 product than to the 18S rRNA mimic. Furthermore, binding to the ITS1 product is as strong as binding to the precursor mimic. Thus, after cleavage, Nob1 is expected to remain bound to the ITS1 product and not to the ribosome product, consistent with previous structure probing and cross-linking analyses [16, 27]. Together, these findings support a model by which Nob1's cleavage at the 3'-end of 18S rRNA promotes its dissociation from the nascent subunit, allowing for subsequent recruitment of mRNAs, thereby setting up a mechanism to ensure only mature subunits enter the translating pool.

On 40S subunits, Nob1 has some steric overlap with the eIF3α subunit of the translation initiation factor eIF3 [19, 22]. eIF3 is essential for recruiting mRNA and the ternary complex to the 40S subunit during translation [49]. Furthermore, the platform region, where Nob1 is located, might also be the site of interaction with the cap-binding complex. These steric conflicts might be the physical reason for the Nob1-dependent block toward mRNA recruitment.

### The Rio1 kinase licenses nascent 40S ribosomes through release of Nob1 and Pno1

Although rRNA cleavage supports the dissociation of Nob1 from nascent ribosomes, it is not sufficient, likely because binding interactions with Pno1 keep it bound to the nascent 40S [18, 22]. Indeed, our genetic and biochemical data demonstrate that Rio1 uses ATP hydrolysis to release both Nob1 and its binding partner Pno1 from nascent ribosomes, thereby regulating

their entry into the translating pool in an ATPase-dependent manner. This role for Rio1 is consistent with previous work in yeast that has shown that Rio1 associates with late pre-40S subunits that retain only Nob1 and Pno1 [27, 28]. In addition, our results are consistent with data from human cells showing that the reimport of Nob1 and Pno1 into the nucleus is more strongly affected by mutations in the Rio1 active site than the reimport of other assembly factors [28].

## How does Rio1 release Nob1 and Pno1?

Analogous to its close relative Rio2, the atypical aspartate kinase Rio1 is believed to use a cycle of autophosphorylation and subsequent dephosphorylation to promote its function in 40S ribosome biogenesis [45, 46]. Our binding data indicate that ATP-bound Rio1 binds ribosome-bound Nob1–Pno1. Furthermore, the release data show that dissociation of the complex requires phosphoryl transfer. We thus speculate that phosphorylation of Rio1 (and presumably release of the ADP) is required to promote a conformational change, which leads to release of Nob1 and Pno1, with the cycle being reset by Rio1 dephosphorylation.

## A quality-control checkpoint is established by Nob1 and Pno1 and regulated by Rio1

Together, the data support a model (Fig 6A) by which the endonuclease Nob1 blocks premature mRNA recruitment. This function is aided by Pno1, which stabilizes Nob1 binding (and also blocks Rps26 recruitment). Because Nob1 release requires rRNA maturation, these two factors set up a mechanism to block the premature release of immature 40S subunits into the translating pool. After Nob1-dependent cleavage of 20S pre-rRNA into mature 18S rRNA, Rio1 releases both Nob1 and Pno1 from nascent 40S subunits, allowing for the recruitment of Rps26 and mRNA and the first round of translation by newly made 40S ribosomes. Thus, Nob1 and Pno1 cooperate to block premature release of immature 40S subunits into the translating pool, and Rio1 regulates the passage through this checkpoint. Whether Rio1 relies simply on the reduced affinity of Nob1 for cleaved rRNA for its temporal regulation or actively recognizes cleaved rRNA will require further studies.

The importance of this safeguard is demonstrated in cells expressing the self-releasing Pno1 mutant Pno1-KKKF, which bypasses Rio1's function and rescues the lethal effect from Rio1 depletion. In these cells, Pno1 can dissociate Rio1 independently, either before or after Nob1 has cleaved the nascent 18S rRNA (Fig 6B). Because Pno1 forms a direct interaction with Nob1 on the pre-40S ribosome [19, 22, 23] and strengthens Nob1's RNA binding affinity [18], its dissociation weakens Nob1, leading to Nob1's release from nascent 40S and 40S recruitment into polysomes. If this spontaneous Pno1 release precedes Nob1 cleavage, then 20S rRNA-containing pre-ribosomes enter the translating pool, where they produce defects in codon selection during translation. Thus, the data herein demonstrate a critical role for Pno1, Nob1, and Rio1 in ensuring only fully matured ribosomes enter the translating pool.

## Why do 20S pre-rRNA-containing ribosomes not support cell growth?

Although the weak-binding Nob1-363 can bypass the requirement for Rio1 in release of rRNA into the polysomes, it does not rescue the cell viability defect from Rio1-deficient cells. Similarly, Nob1-deficient yeast accumulate ribosomes that can translate but not support cell viability. The data herein show that in both cases these immature ribosomes retain Pno1, thus preventing the incorporation of the essential protein Rps26 [19, 20].

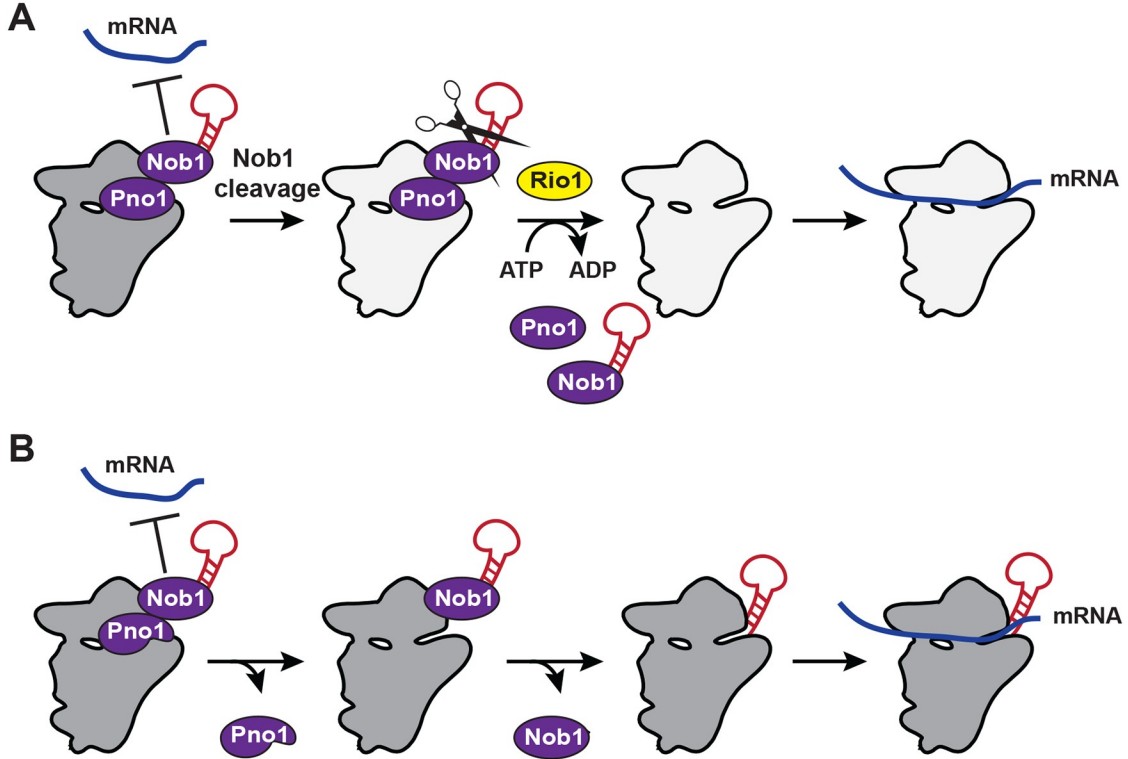

**Fig 6. Model of Rio1's role in 40S ribosome biogenesis.** (A) Nob1 blocks mRNA recruitment. Following Nob1-mediated cleavage of 20S pre-rRNA to form the mature 18S rRNA, Rio1 binds the 40S-bound Nob1–Pno1 complex and releases both assembly factors from the nascent ribosome in an ATPase-dependent manner. This allows for Rps26 incorporation and mRNA recruitment. Therefore, Rio1 regulates the final stages of 40S maturation in the cytoplasm to only release fully mature ribosomes into the translating pool. (B) When Rio1 is bypassed, such as in cells expressing the weak-binding Pno1-KKKF, Pno1 can dissociate Rio1 independently, even before Nob1-dependent rRNA cleavage. Because Pno1 is required for D-site cleavage [18], this reduces the endonuclease activity of Nob1. Eventually Nob1 can be released, even when RNA cleavage has not occurred, allowing mRNA recruitment by immature 20S pre-rRNA-containing 40S ribosomes.

Rps26 depletion leads to accumulation of 20S rRNA [50], suggesting that Rps26 would be incorporated prior to rRNA maturation, not after, as suggested by the data herein. These observations can be reconciled by the surprise finding that fully matured 18S rRNA-containing, but Rps26-depleted, ribosomes do not efficiently translate the late 40S assembly factor Fap7 [42]. Thus, Rps26 depletion affects 20S maturation indirectly by blocking the production of Fap7, leading to the accumulation of 20S rRNA-containing 40S assembly intermediates. This model, which reconciles the data herein with the data from Schutz and colleagues, is further supported by the observation that high-copy Fap7 is a suppressor of reduced amounts of Rps14 [51].

## Other cellular roles for Rio1

Previous work has also established roles for Rio1 in cell division, in which it binds ribosomal DNA (rDNA) and interacts with the regulator of nucleolar silencing and telophase exit (RENT) complex and the helicase Sgs1, which both regulate rDNA silencing, the stability of the locus, and its condensation during cell segregation. Accordingly, Rio1-depleted cells have defects in these processes [52, 53]. The rescue of cell viability in the absence of Rio1 by the Pno1-KKKF mutant indicates that the essential role of Rio1 is in ribosome assembly. However,

the small but significant difference in doubling times of cells containing Pno1-KKKF with and without Rio1 (S5B Fig) would be consistent with additional (nonessential) roles for Rio1 outside of ribosome assembly.

## Bypassing the Rio1 checkpoint disturbs protein homeostasis and may promote cancer

Rio1 is conserved throughout all domains of life and plays an important role during ribosome assembly in human cells [31]. Intriguingly, whole-genome sequencing of cancer cells reveals that diverse cancers accumulate mutations in Pno1 that are either directly adjacent to Pno1-KKKF or similarly contact either the rRNA, Nob1, or ribosomal proteins (S7 Fig, TCGA Research Network: https://www.cancer.gov/tcga). Thus, although it remains unclear whether these mutations play any role in promoting cancer progression, like Pno1-KKKF, the cancer-associated Pno1 mutants are expected to bypass Rio1, leading to the release of immature ribosomes into the translating pool and resulting in translation fidelity defects, as we have shown in yeast cells.

## Materials and methods

### Yeast strains and cloning

*Saccharomyces cerevisiae* strains used in this study were obtained from the GE Dharmacon Yeast Knockout Collection or were made using PCR-based recombination [54]. Strain identity was confirmed by PCR and western blotting when antibodies were available. Mutations in plasmids were made by site-directed mutagenesis and confirmed by sequencing. Rio1 was cloned into pSV272 for expression as a TEV-cleavable His$_6$-MBP fusion protein. Plasmids were propagated in XL1 Blue competent cells. Yeast strains and plasmids used in this study are listed in S1 and S2 Tables, respectively.

### Protein expression and purification

Pno1, MBP-Pno1, Nob1, and MBP-Nob1 were purified as previously described [16, 18, 55]. Truncated Nob1-363 was purified using the same protocol as the WT protein.

To express and purify Rio1, Rosetta DE3 competent cells transformed with a plasmid encoding His-MBP-tagged Rio1 were grown to mid-log phase at 37 ˚C in LB media supplemented with the appropriate antibiotics. Rio1 expression was induced by addition of 1 mM isopropyl β-D-thiogalactoside (IPTG), and cells were grown for another 5 h at 30 ˚C. Cells were lysed by sonication in Ni-NTA lysis buffer supplemented with 0.5 mM phenylmethylsulfonyl fluoride (PMSF) and 1 mM benzamidine. The cleared lysate was purified over Ni-NTA affinity resin according to the manufacturer's recommendation (Qiagen). Eluted proteins were pooled and dialyzed overnight at 4 ˚C into 50 mM Na$_2$HPO$_4$ (pH 8.0), 150 mM NaCl, and 1 mM DTT. Protein was applied to a MonoQ column in the same buffer and eluted with a linear gradient of 150 mM to 600 mM NaCl over 12 column volumes. The protein was pooled and concentrated for further purification on a Superdex200 size-exclusion column equilibrated with (50 mM HEPES [pH 8.0], 200 mM NaCl, 1 mM DTT, 1 mM TCEP). Protein concentration was determined by absorption at 280 nm using an extinction coefficient of 106,120 $M^{-1}cm^{-1}$.

Untagged Rio1 was purified as described above, except that 0.76 μg/mL TEV protease was added during dialysis. Protein concentration was determined by absorption at 280 nm using an extinction coefficient of 36,790 $M^{-1}cm^{-1}$. Rio1-D261A and Rio1-D244A were purified using the same protocol as the WT protein.

## Sucrose density gradient analysis

Sucrose gradient fractionation of whole-cell lysates followed by northern blot analysis were performed as described previously [5]. Briefly, cells were grown to mid-log phase in the appropriate media (indicated in the respective figure legends), harvested in 0.1 mg/mL cycloheximide, washed, and lysed in gradient buffer (20 mM HEPES [pH 7.4], 5 mM $MgCl_2$, 100 mM KCl, and 2 mM DTT) with 0.1 mg/mL cycloheximide, complete protease inhibitor cocktail (Roche), 1 mM benzamidine, and 1 mM PMSF. Cleared lysate was applied to 10%–50% sucrose gradients and centrifuged in an SW41Ti rotor for 2 h at 40,000 RPM and then fractionated. The percent of 20S pre-rRNA in the polysomes was calculated by dividing the amount of 20S pre-rRNA in the polysome fractions (fractions 8–13) by the total amount of 20S pre-rRNA in all fractions (fractions 2–13).

## Northern analysis

Northern blotting was carried out essentially as previously described [6], using the following probes: 20S, GCTCTCATGCTCTTGCC; 18S, CATGGCTTAATCTTTGAGAC; 25S, GCCCGTTCCCTTGGCTGTG; and U2, CAGATACTACACTTG.

## Protein binding assays

In total, 7 μM of MBP-tagged protein (MBP-Rio1, MBP-Pno1, and MBP-Nob1) was mixed with 20 μM untagged protein (Rio1, Nob1, or Pno1) in binding buffer (50 mM HEPES [pH 7.5], 200 mM NaCl, and 5 mM $MgCl_2$). In all, 2 mM ATP, ADP, or AMPPNP was added where indicated. Proteins were preincubated at 4 ˚C for 30 min before addition of 100 μL equilibrated amylose resin (New England BioLabs). The mixture was incubated for 1 h at 4 ˚C, the flow through was collected, the resin was washed with binding buffer supplemented with 0.8 mM ATP, ADP, or AMPPNP where indicated, and proteins were eluted with binding buffer supplemented with 50 mM maltose.

## RNA binding assay

RNA binding assays were performed as previously described [16]. Briefly, [32]P-ATP-labeled H44-A2, H44-D, or D-A2 RNAs, named after the sequence on structural elements that mark their start and end points, were prepared by transcription in the presence of α-ATP, gel purified, and eluted via electroelution. These RNAs have been validated to fold into well-defined structures relevant to ribosome assembly [16, 56]. RNAs were then precipitated and resuspended in water. RNAs were folded by heating for 10 min at 65 ˚C in the presence of 40 mM HEPES (pH 7.6), 100 mM KCl, and 2 mM $MgCl_2$. Trace amounts of radiolabeled RNA were incubated with varying concentrations of Nob1 in 40 mM HEPES (pH 7.6), 50 mM KCl, and 10 mM $MgCl_2$ for 10 min at 30 ˚C. Samples were loaded directly onto a running 6% acrylamide/THEM native gel to separate protein-bound from unbound RNAs. After drying the gel, phosphorimager analysis was used to quantify the gel. Bound RNA was plotted against protein concentration and fit with a single binding isotherm to obtain apparent binding constants using KaleidaGraph version 4.5.4 from Synergy Software.

## Release assay

Pre-ribosomes from Rio1-depleted cells were purified from Gal::Pno1; Gal::Rio1 cells transformed with a plasmid encoding TAP-Pno1 and grown in YPD medium for 16 h essentially as described before [6]. In all, 40 nM of pre-40S ribosomes were incubated with 2 μM purified, recombinant Rio1, Rio1-D244A, or Rio1-D261A in 50 μL of buffer (50 mM Tris-HCl [pH 7.5],

100 mM NaCl, 10 mM MgCl$_2$, 0.075% NP-40, 0.5 mM EDTA, and 2 mM DTT). ATP, AMPPNP, or ADP was added to a final concentration of 1 mM. The samples were then incubated at room temperature for 10 min, placed on 400 μL of a 20% sucrose cushion, and centrifuged for 2 h at 400,000$g$ in a TLA 100.1 rotor. The supernatant was TCA-DOC precipitated, and the pellets were resuspended in SDS loading dye. Supernatants (released factors) and pellets (bound factors) were analyzed by SDS-PAGE followed by western blotting.

### Quantitative growth assays

Stress-tolerance tests were performed as previously described [42]. In brief, Gal::Nob1 cells transformed with a plasmid encoding Nob1 or an empty vector were grown to mid-log phase in galactose dropout media, switched to glucose dropout media for 10 h and grown to mid-log phase, and then inoculated into stress media (or control cultures) at OD 0.05 to test stress tolerance. The stress medium was either YPD + 1 M NaCl (high salt) or 10 mM caffeine, and YPD was used as the control medium.

To measure the doubling times of cells expressing Pno1 or Pno1-KKKF with and without Rio1, Gal::Pno1; Gal::Rio1 cells transformed with a plasmid encoding Pno1 or Pno1-KKKF and a second plasmid encoding Rio1 or an empty vector were grown to mid-log phase in glucose dropout medium for 20 h to deplete endogenous Pno1 and Rio1 and then inoculated into the same medium at OD 0.05. Cells were grown at 30 ˚C while shaking, and the doubling times were measured in a Synergy 2 multimode microplate reader (BioTek).

### Dual-luciferase reporter assay

Gal::Nob1 cells grown in glucose media were supplemented with either WT Nob1 or an empty vector. Cells were harvested in mid-log phase, and reporter assays were carried out essentially as described before [6]. Cells were lysed, and luciferase activity was measured with the Promega Dual-Luciferase Reporter Assay System on a PerkinElmer EnVision 2104 Multilabel Reader according to the manufacturer's protocol, with assay volumes scaled down to 15%. For each sample, firefly luciferase activity was normalized against renilla activity; subsequently, values observed for depleted Nob1 were normalized against those for WT Nob1.

### Antibodies

Antibodies against recombinant Nob1, Pno1, and Rps10 were raised in rabbits by Josman or New England Peptide and tested against purified recombinant proteins and yeast lysates. Antibody against phosphor-eIF2α was purchased from Thermo Fisher Scientific (Cat# 44-728G).

### ATPase assay

In total, 10 μM purified, recombinant Rio1, Nob1, and Pno1 were incubated with trace amounts of $^{32}$P-ATP in ATPase buffer (10 mM MgCl$_2$, 100 mM KCl, 500 mM HEPES [pH 7.5]) at 30 ˚C for the indicated times. At each time point, the reactions were stopped in quench buffer (0.75 M KH$_2$PO$_4$ [pH 3.3]). The samples were run on a TLC PEI Cellulose F plate using the quench buffer as the solvent. Phosphorimager analysis was used to quantify the TLC plate. The fraction of hydrolyzed ATP was plotted against incubation time at 30 ˚C.

### Quantification and statistical analysis

Quantification of northern blots and ATPase assays was performed using Quantity One 1-D Analysis Software version 4.1.2, and quantification of western blots was performed using Image Lab version 5.2.1, both from Bio-Rad Laboratories. Statistical analysis of the dual-

luciferase translation fidelity assay was performed using GraphPad Prism version 6.02 (Graph-Pad Software, La Jolla, California, United States, www.graphpad.com). Statistical analyses of northern blots and growth assays were performed using the programming language R in Rstudio, version 3.2.3 (https://www.R-project.org/). Samples grown and analyzed on the same day were considered paired replicates, and significance was calculated using a paired, two-tailed $t$ test. Otherwise, an unpaired, two-tailed $t$ test was used as indicated in the figure legends.

## Supporting information

**S1 Fig. Rio1 depletion and the Nob1-D15N mutation result in a similar phenotype (related to Figs 1 and 3).** (A) Depletion of Nob1 or expression of Nob1-D15N does not induce the cellular stress response. Western blots from total yeast cell lysates, probed for the indicated proteins. (B) Growth of WT yeast cells transformed with an e.v. or Nob1 or Nob1-D15N under the galactose-inducible, glucose-repressible Gal promoter were compared by 10-fold serial dilutions on glucose or galactose dropout plates. (C) Northern blot analyses of total cellular RNA from cells depleted of Nob1 grown in glucose for the indicated times and total cellular RNA from WT BY4741 cells overexpressing Nob1-D15N or transformed with an e.v. grown in galactose for the indicated times. (D) Shown are 10%–50% sucrose gradient from cell lysate of Tsr1-TAP; Gal:: Rio1 cells depleted of Rio1 by growth in YPD for 16 h. Northern blots of 20S, 18S, and 25S rRNA and western blots probing for Nob1 and Pno1 are shown below the absorbance profile at 254 nm. Arrowheads note the bands corresponding to Nob1 and Pno1. Most 20S rRNA accumulated in 80S-like ribosomes (fraction 6). e.v., empty vector; WT, wild-type. (TIF)

**S2 Fig. Only overexpression of Rio1 rescues the dominant-negative phenotype of the Nob1-D15N mutation (related to Fig 3).** Growth of the indicated cells containing an empty vector or Nob1 or Nob1-D15N under the Gal promoter were compared by 10-fold serial dilutions on galactose or glucose dropout plates. Gal, galactose. (TIF)

**S3 Fig. Rio1 does not affect Nob1-depleted cells or wild-type cells (related to Fig 3).** (A) Overexpression of Rio1 does not rescue Nob1 depletion. Growth of cells containing Nob1 under a Gal promoter and expressing either Nob1 or Rio1 from a plasmid under a copper-inducible (Cup1) promoter or an empty vector were compared by 10-fold serial dilutions on glucose or galactose dropout plates with 100 μM CuSO$_4$. (B, C) Sucrose gradient from wild-type cells transformed with an empty vector and overexpressing wild-type Nob1 under a Gal promoter grown in galactose with 100 μM CuSO$_4$ for 16 h. Shown below the absorbance profile at 254 nm are northern blots of 20S, 18S, and 25S rRNAs and western blots probing for Nob1 and Pno1. Arrowheads note the bands corresponding to Nob1 and Pno1. Gal, galactose. (TIF)

**S4 Fig. Rio1 does not bind Nob1 or Pno1 individually (related to Fig 4).** (A) Rio1 does not bind Nob1 or Pno1 individually. Shown are Coomassie-stained SDS-PAGE gels of protein binding assays of purified, recombinant MBP-Rio1, Rio1, MBP-Nob1, Nob1, MBP-Pno1, and Pno1 in the presence of AMPPNP. (B) Coomassie-stained SDS-PAGE gels of protein binding assays on amylose beads of purified, recombinant MBP-Nob1, Nob1, MBP-Pno1, Pno1, and Rio1 in the presence of AMPPNP or ADP. The order of the samples was edited for clarity. (C) Rio1 does not bind MBP. Shown is a Coomassie-stained SDS-PAGE gel of a protein binding assay of purified, recombinant MBP and Rio1. Nob1 and Pno1 also do not bind MBP alone [18]. *MBP. (D) Addition of Nob1 and Pno1 (squares) does not increase the rate of ATP hydrolysis by Rio1 (circles). Numerical data are listed in S1 Data. AMPPNP, adenylyl-

imidodiphosphate; E, elution; FT, flow through; In, input; MBP, maltose-binding protein; W, final wash.
(TIF)

**S5 Fig. Rescue of the Rio1 depletion phenotype is specific to Pno1-KKKF (related to Fig 4).** (A) Growth of cells expressing wild-type Pno1 or Pno1 mutants with and without Rio1 were compared by 10-fold serial dilutions on glucose and galactose dropout plates. Pno1-GXXG (N111G/S112K/W113D/T114G), Pno1-WK/A (W113A/K115A), Pno1-HR/E (H104E/R105E), Pno1-DDD/K (D167K/D169K/D170K). (B) Quantitative growth measurements for cells expressing Pno1 or Pno1-KKKF in the presence or absence of Rio1. Five biological replicates, error bars represent SEM, and $^{****}p < 0.0001$ via unpaired $t$ test. Numerical data are listed in S1 Data. (C) Growth of cells expressing wild-type Nob1 or Nob1 mutants with or without Rio1 were compared by 10-fold serial dilutions on glucose and galactose dropout plates. (D) Growth of cells containing endogenous Rio1 under a Gal promoter expressing either wild-type Nob1 or Rio1 under a copper-inducible (Cup1) promoter or an empty vector were compared by 10-fold serial dilutions on glucose or galactose dropout plates with 100 μM CuSO$_4$. Gal, galactose.
(TIF)

**S6 Fig. Truncated Nob1-363 weakly binds RNA (related to Fig 5).** (A) Growth of cells expressing wild-type Nob1 or Nob1 mutants under the Tef2 or Cyc1 promoter, as indicated, with or without Rio1 were compared by 10-fold serial dilutions on glucose and galactose dropout plates. The Tef2 promoter produces higher protein levels [57]. (B) RNA binding assay with in vitro transcribed H44-A2 RNA (20S pre-rRNA mimic) and recombinant Nob1 or Nob1-363. Three independent experiments yielded values of $K_d = 0.37 +/− 0.22$ μM for Nob1 binding H44-A2 (white circles) and $K_d = 1.43 +/− 0.22$ μM for Nob1-363 binding H44-A2 (white squares). Numerical data are listed in S1 Data.
(TIF)

**S7 Fig. Pno1 mutations discovered in cancer genomes (related to the Discussion).** Mutations in Pno1 that accumulate in diverse cancers (green space fill, from the TCGA Research Network: https://www.cancer.gov/tcga) are directly adjacent to Pno1-KKKF (yellow space fill) or similarly contact either the rRNA, Nob1, or ribosomal proteins. Premature 18S rRNA (from human pre-40S, surface view in grey) bound by Nob1 (cyan) and Pno1 (magenta). Image was obtained from PDB 6G18 (human pre-40S state C, [22]). For simplicity, all proteins other than Nob1 and Pno1 are omitted. TCGA, The Cancer Genome Atlas.
(TIF)

**S1 Table. Yeast strains used in this work [5,58].**
(DOCX)

**S2 Table. Plasmids used in this work.**
(DOCX)

**S1 Data. Excel spreadsheet containing the numerical values for each of the graphs represented in the manuscript.** This file has individual tabs for each Figure.
(XLSX)

**S1 Raw Images. This file contains the uncropped images of western and northern gels and gel shift assays in the manuscript.**
(PDF)

## Acknowledgments

We thank A. Lamanna for the gift of recombinant Nob1-363 and G. Dieci, J. Warner, A. G. Hinnebusch, and T. E. Dever for gifting us the anti-Rps8, anti-Rpl3, anti-eIF1, and anti-eIF2α antibodies, respectively. The dual-luciferase plasmids were kindly provided by J. Lorsch (start site recognition), D. Bedwell (stop codon read through and miscoding), and J. Dinman (frame-shifting). We thank T. Mueller and members of the Karbstein lab for discussion and comments on the manuscript.

## Author Contributions

**Conceptualization:** Melissa D. Parker, Katrin Karbstein.

**Formal analysis:** Melissa D. Parker, Jason C. Collins, Boguslawa Korona, Homa Ghalei, Katrin Karbstein.

**Funding acquisition:** Katrin Karbstein.

**Investigation:** Melissa D. Parker, Jason C. Collins, Boguslawa Korona, Homa Ghalei.

**Project administration:** Katrin Karbstein.

**Resources:** Melissa D. Parker, Boguslawa Korona, Homa Ghalei.

**Supervision:** Katrin Karbstein.

**Writing – original draft:** Melissa D. Parker, Katrin Karbstein.

**Writing – review & editing:** Melissa D. Parker, Jason C. Collins, Katrin Karbstein.

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
