## [Editor Report · Decision Letter 0]

30 May 2019

Dear Dr Karbstein, 

Thank you for submitting your manuscript entitled "A kinase-dependent checkpoint prevents escape of immature ribosomes into the translating pool" for consideration as a Research Article by PLOS Biology.

Your manuscript has now been evaluated by the PLOS Biology editorial staff as well as by an academic editor with relevant expertise and I am writing to let you know that we would like to send your submission out for external peer review.

Please re-submit your manuscript within two working days, i.e. by Jun 01 2019 11:59PM.

Kind regards,

Hashi Wijayatilake, PhD,

Managing Editor

PLOS Biology

---

## [Decision Letter · Decision Letter 1]

8 Jul 2019

Dear Dr Karbstein,

Thank you very much for submitting your manuscript "A kinase-dependent checkpoint prevents escape of immature ribosomes into the translating pool" for consideration as a Research Article at PLOS Biology. Your manuscript has been evaluated by the PLOS Biology editors, an Academic Editor with relevant expertise, and by three independent reviewers.

In light of the reviews (below), we will not be able to accept the current version of the manuscript, but we would welcome resubmission of a much-revised version that takes into account the reviewers' comments. Our academic editor advises that reviewer 1’s request for Rio1 ATPase measurements +/- Nob1-Pno1 is reasonable and would provide support for your model that “Rio1 releases Nob1 and Pno1 from nascent ribosomes in an ATPase- dependent manner”. Reviewer 3 also requests additional evidence to support this claim. Specifically, you are suggested to repeat the Nob1/Pno1 ribosome release assays comparing WT Rio1 with kinase and ATP-binding mutants. Our academic editor believes that this is a good and reasonable suggestion. Reviewer 2 raises concern about the evidence for suppression of Rio1 depletion by Pno1-KKKF and requests an alternative genetic approach that is feasible. Reviewer 3 raises related concerns about the characterization of the Pno1-KKKF strain by semi-quantitative western blotting for various RPs. The proposed quantitative mass spec analysis is an expensive experiment, but one you have used previously in related work, so this suggestion is not unreasonable. Our academic editor agrees with reviewer 3 that the western blots are hard to interpret and suggests that relating the genetic interaction (between Rio1 depletion and Pno1-KKKF) to a molecular phenotype is important for the story; this is the main evidence supporting your claim that “bypassing Rio1 via self-releasing mutations in Pno1 results in release of immature ribosomes” into the translating pool where they malfunction. Our academic editor indicates that although reviewer 3 proposes many additional experiments, most are presented as optional if you modify the text to remove strong claims from weak evidence. We are inclined to let you decide which of these experiments you want to do, and let the reviewer decide whether the text has been modified adequately in the revision.

We cannot make any decision about publication until we have seen the revised manuscript and your response to the reviewers' comments. Your revised manuscript is also likely to be sent for further evaluation by the reviewers.

Your revisions should address the specific points made by each reviewer. Please submit a file detailing your responses to the editorial requests and a point-by-point response to all of the reviewers' comments that indicates the changes you have made to the manuscript. In addition to a clean copy of the manuscript, please upload a 'track-changes' version of your manuscript that specifies the edits made. This should be uploaded as a "Related" file type. You should also cite any additional relevant literature that has been published since the original submission and mention any additional citations in your response. 

Before you revise your manuscript, please review the following PLOS policy and formatting requirements checklist PDF: http://journals.plos.org/plosbiology/s/file?id=9411/plos-biology-formatting-checklist.pdf. It is helpful if you format your revision according to our requirements - should your paper subsequently be accepted, this will save time at the acceptance stage.

Please note that as a condition of publication PLOS' data policy (http://journals.plos.org/plosbiology/s/data-availability) requires that you make available all data used to draw the conclusions arrived at in your manuscript. If you have not already done so, you must include any data used in your manuscript either in appropriate repositories, within the body of the manuscript, or as supporting information (N.B. this includes any numerical values that were used to generate graphs, histograms etc.). For an example see here: http://www.plosbiology.org/article/info%3Adoi%2F10.1371%2Fjournal.pbio.1001908#s5.

For manuscripts submitted on or after 1st July 2019, we require the original, uncropped and minimally adjusted images supporting all blot and gel results reported in an article's figures or Supporting Information files. We will require these files before a manuscript can be accepted so please prepare them now, if you have not already uploaded them. Please carefully read our guidelines for how to prepare and upload this data: https://journals.plos.org/plosbiology/s/figures#loc-blot-and-gel-reporting-requirements.

Upon resubmission, the editors will assess your revision and if the editors and Academic Editor feel that the revised manuscript remains appropriate for the journal, we will send the manuscript for re-review. We aim to consult the same Academic Editor and reviewers for revised manuscripts but may consult others if needed.

We expect to receive your revised manuscript within two months. Please email us (plosbiology@plos.org) to discuss this if you have any questions or concerns, or would like to request an extension. At this stage, your manuscript remains formally under active consideration at our journal; please notify us by email if you do not wish to submit a revision and instead wish to pursue publication elsewhere, so that we may end consideration of the manuscript at PLOS Biology.

When you are ready to submit a revised version of your manuscript, please go to https://www.editorialmanager.com/pbiology/ and log in as an Author. Click the link labelled 'Submissions Needing Revision' where you will find your submission record. 

Sincerely,

Di Jiang, PhD

Associate Editor

on behalf of 

Hashi Wijayatilake, PhD, 

Managing Editor

PLOS Biology

Reviewer remarks:

Reviewer #1: In this manuscript Parker et al utilize yeast genetics, cell based assays, and in-vitro RNA binding and protein binding assays to further our understanding of the role of three essential ribosome assembly factors Nob1, Pno1, and Rio1 in the assembly of the small ribosomal subunit. Overall the work is novel, comprehensive, well done and supports the model that the Rio1-Nob1-Pno1 network safeguards cells from releasing immature 40S subunits into the pool of translating ribosomes. This manuscript is worthy of publication in PLOS Biology however I have a few concerns that need to be addressed. 

1. I disagree with the authors interpretation of Fig. 2. The difference in kd for the three different RNA substrates is very small. While I agree that binding is weaker once A2 is gone, I don’t understand how the authors can claim this signifies that Nob1 dissociates with A2 following cleavage. To truly make this type of claim the authors should carry out competitive binding assay with H44 and A2. Moreover as the authors point out in the manuscript cleavage alone is not sufficient for Nob1 release.

2. The authors nicely show that Nob1-Pno1 interact with Rio1 in an ATP dependent manner and that Rio1 can release Nob1-Pno1 from nascent 40S subunits in vitro. Together this data suggests that Nob1-Pno1 stimulates Rio1 ATP hydrolysis. The authors should measure rates of Rio1 ATP hydrolysis in the presence and absence of Nob1-Pno1. It would be nice but it is not essential if the authors could show that a Walker B mutant of Rio1 blocks Nob1-Pno1 release in their in vitro release assay. 

3. There appears to be some inconsistency with the Nob1 Western Blots. In contrast the Pno1 Westerns are much more consistent. Some Nob1 blots show a high degree of non-specificity and it’s unclear how the authors know which band corresponds to Nob1. For example in the Westerns in Figure S1, B has a single band, C has 3 bands and D has 2 bands. The exposure time of the Nob1 westerns also varies quite a bit from figure to figure. The authors should either repeat these westerns, provide an explanation (protein degradation perhaps?), and/or include the uncropped western blots as a supplemental figure. 

Minor Concerns:

1. In Fig. 4A the contrast level of the gel is too high, making it very hard to see the band for Pno1. 

2. The model in Fig. 6 is a little hard to follow. I would suggest that the authors split the top and bottom into two separate panels, so they can distinguish WT-Pno1 from Pno1-KKKF.

Reviewer #2: The work by Parker and coworkers establishes an important quality control checkpoint of the final step of 40S subunit biogenesis. The work focusses on the role of the nuclease Nob1, required for the processing of 20S to 18S rRNA, the Nob1 binding partner Pno1, and the atypical kinase Rio1. Involvement of these factors in 40S maturation was previously reported. The authors here confirm some of the earlier findings, e.g. that Rps26 is assembled into 40S subunits late, after removal of Pno1. In addition the work provides novel insight into how the last steps of 40S maturation allow for proper quality control. The data reveal that 20S rRNA processing is required for the release of Pno1 and Nob1 from pre-40S particles. The data show that Rps26 only assembles after Pno1 release and rRNA processing, by that marking 40S subunits as "functional". In addition the release of Pno1/Nob1 requires the activity of Rio1 kinase. Interestingly, Rio1 requirement (Rio1 is normally essential) can be overcome my mutations within Pno1, which destabilize ribosome-binding of Pno1. The data reveal the such mutations within Pno1 result in the production of translating ribosomes, containing 20S rRNA and lacking Rps26. The data further suggest that Rio1-independent release of Pno1 mutants, with ribosome-binding defects rescues the lethality of Rio1. 

The presented data are mostly convincing, clearly presented, and justify the conclusions drawn. I have a few comments for the authors.

My only major concern is that suppression of Rio1-depletion by Pno1-KKKF is not that convincing. In Fig. 4C is looks fine, however, in Fig. S3a I can hardly see a difference e.g. between the effect of Pno1-KKKF and Pno1-DDD/K (and also Pno1 wild type ...).The GAL depletion system has its problems. I suggest to show rescue in a more stringent way: use a diploid heterozygous Drio1Dpno1 strain, transform with the wild type Pno1 or the relevant mutants on a plasmid, dissect, and show that the strain Drio1Dpno1 + plasmid-borne Pno1-KKKF is viable. This is doable and would significantly strengthen one of the major conclusions of this work. 

In this context. Rio1 was found to act also in other cellular processes. This should be discussed also with respect to the complementation of Drio1 by Pno1-KKKF. Do the authors think that the 40S maturation function is the only essential function of Rio1? 

Page 6: The observed differences with respect to the Kds are rather moderate. I would not over interpret these data.

Page 7: Please indicate what is ment by a "limited screen" and which factors were tested in this screen.

Page 8: "Rio1 phosphorylation functions as a switch, which is reset after hydrolysis". I did not understand this sentence. Please include 2-3 sentences, explaining the current model (or the model of the authors!) of how Rio1 is supposed to affect release of Pno1/Nob1. Rio1 seems to bind in the ATP-bound state. And then? In this context: "kinase-dependent" (in the Title of the work) to the general reader suggests that some phosphorylation event is involved in the process. Is this the idea? Or do the authors think that it is rather ATP hydrolysis by which Rio1 drives the process? Please provide some mechanistic ideas (even if speculative). 

The work by Schutz et al 2014 suggested that Rps26 is required for cytoplasmic processing of 20S pre-rRNA to mature 18S rRNA. Please discuss these findings in the light of your data. 

The probes used to detect 18S and 20S rRNA species should be given in the Supplemental material.

Reviewer #3: Parker et al. use yeast genetics and biochemical reconstitution to provide evidence for how three late-stage 40S ribosome assembly factors (AFs) coordinate to establish a quality-control checkpoint that prevents release of immature 40S subunits into the active translational pool. While the study provides an exciting new model with some compelling data, the challenges for the study are the methods used for quantitation in many places (“quantitated” western blots which are notoriously problematic) and the relatively modest magnitude of effects. The strengths of the study are the use of yeast genetics as a method to identify roles for critical factors and the coupling with ribosome biochemistry. The section at the end focusing on translational defects associated with the RPS26-deficient ribosomes was particularly weak and added no mechanistic insight. Overall, the authors do provide new potential insights into a Rio1- and Nob1-mediated checkpoint that prevents release of immature ribosomes into the translating pool, though more rigorous quantitative approaches are needed to support this new model. 

Systematic comments:

The authors initially show that in yeast depleted of the endonuclease Nob1, immature 40S subunits containing unprocessed 20S rRNA (and Nob1’s binding partner, Pno1) enter the active translational pool. However, a catalytically inactive dominant-negative Nob1 (D15N) prevents escape of such immature ribosomes, suggesting that Nob1 blocks mRNA recruitment and entry of immature ribosomes into the translational pool. This is a striking and interesting lead for the manuscript.

Comment: This result is initially confusing and could be better discussed at this early stage – if Pno1 is present then these immature ribosomes presumably lack RPS26 and yet are still translating (this connects with earlier work from this laboratory). Given this striking phenotype, this would be an appropriate place to mention that while these immature ribosomes translate, they most likely suffer from translational defects (as eventually characterized by reporter assays, Fig 5E). Additionally, these are clearly very sick cells where few polysomes are formed. Do the authors check whether ribosome levels, global protein amounts and/or homeostasis is perturbed in �Nob1 strains? Is the UPR (characterized by eIF2� phosphorylation) upregulated in �Nob1 strains? If so, can the phenotype be rescued by Nob1-D15N expression in the �Nob1 background?

Next, the authors hypothesize that cleavage of rRNA by Nob1 is a prerequisite for its release from immature 20S-rRNA-containing ribosomes. To address this question, the authors use gel-shift assays to show that Nob1 binds its 20S substrate-mimic (H44-A2) and 3’ cleavage product (D-A2) more efficiently compared to the 18S mature rRNA product-mimic (H44-D). 

Comment: The binding affinities reported for all substrates are in the high nM (low µM) range which seems weak for typical RNA-protein interactions. More importantly, there is a modest two-fold difference in binding affinity between the substrate (H44-A2) and mature product (H44-D) mimics. Given this modest difference in affinity, the authors should tone down their statements when stating differences in these relative binding affinities (Page 13 “…binding affinities for the precursor rRNA and ITS1 product are indistinguishable, and much stronger than for the mature 18S rRNA product.”

Next, the authors performed a clever overexpression (OE) screen to identify candidates that rescue the dominant-negative growth phenotype of Nob1-D15N and identified the kinase Rio1 as rescuing the D15N growth phenotype; the growth phenotype was very convincing for this experiment. The authors go on to try to show that Rio1 OE rescues the D15N phenotype by releasing immature 20S pre-rRNA containing ribosomes into the translating pool by quantitating 20S rRNA in the fractions across a sucrose gradient. 

Comment: The RNA analysis for Figure 3 was not compelling. While the authors are clearly able to run gels that nicely distinguish between 18S and 20S RNAs, these products were not sufficiently resolved in Figure 3B, nor was the decision to quantify relative to U2 totally obvious to me. The sucrose gradient is the better experiment, but here the effects are relatively modest (30% vs. 53%) though the exposures seem to have been deliberately chosen to make the result look stronger than the quantitation reveals. Fundamentally, the dominant negative is not that potent, and so the background is high for the experiment. At a minimum, these experiments should be performed such that the RNA products are well resolved. 

The authors performed pulldown assays to show that MBP-Rio1 binds the Nob1•Pno1 complex in the presence of non-hydrolysable ATP. 

Comment: While these experiments are well-controlled showing that Rio1 specifically binds the Nob1•Pno1 complex, but not Nob1 and Pno1 individually, based on the data it is impossible to interpret/conclude whether Rio1 recognizes the Nob1•Pno1 interface – the pulldowns in Figure 4A clearly show that Nob1 and Pno1 are not stoichiometric - the conclusion is overstated. If the authors wish to claim that Rio1 recognizes the Nob1•Pno1 interface, they should perform cross-linking MS analyses to determine residues at the binding interface based off distance constraints imposed by the cross-linker. 

The authors set up a pelleting assay using ribosomes purified from TAP-tagged Pno1-�Rio1 strains to show that addition of recombinant Rio1 and ATP results in release of Nob1 and Pno1 from ribosomes. 

Comment: Although these are likely difficult experiments, the release phenotypes for Nob1 and Pno1 with Rio1 and ATP are very weak (this could stem from suboptimal reaction conditions, incomplete reaction time-course, or limited Rio1 enzymatic activity). Moreover, western blots are semi-quantitative and so this experiment does not strongly support the prediction that Rio1 dissociates Nob1 and Pno1. The results further indicate that ~20% Nob1 is released the presence of ADP (compared to 40% with ATP). Since the background is again high for this experiment, the authors should compare WT- Rio1 with the kinase and ATP-binding mutants to test if they can get a stronger phenotype.

Next, the authors screened for weak-ribosome binding mutants for Nob1 and Pno1 that rescue the growth-defect imposed by �Rio1, and identified Pno1-KKKF and Nob1-1-363 as mutants/deletions that rescue the �Rio1 phenotype and enable immature ribosomes to enter the translational pool. The authors further hypothesize that the immature 20S rRNA and Pno1-containing ribosomes would lack RPS26 due to binding-interface incompatibility, and therefore tested whether RPS26 is sub-stoichiometric in ribosomes purified from Pno1-KKKF strains (compared to Pno1-WT strains). To perform this analysis, they chose 5 different r-proteins for normalization.

Comment: While RPS26 appears to be sub-stoichiometric compared to RPS0 and RPS2 in Pno1-KKKF strains, the ratio is identical when comparing RPS5, RPS10 and RACK1. The authors should elaborate on why they chose the specific RPs for comparison? As before, westerns blots are semi-quantitative making it hard to interpret the result. A better experiment would involve isolating polysomes from KKKF and WT strains, normalizing for equal ribosomal amounts, and quantitative mass-spectrometry. From this data, they could interpret whether the ratio of RPS26 compared to an array of other RPs is lower in the KKKF compared to WT strains. Normalizing for total ribosomes is important, because from the sucrose gradient profiles it is evident that Pno1-KKKF has increased 60S and decreased polysome populations, suggesting a global defect in ribosomal numbers. 

In the final section, the authors attempt to connect the S26 deficiency to earlier work from their group, arguing that these immature S26-deficient ribosomes are stress-tolerant and translationally perturbed using growth assays and luciferase reporters. 

Comment: The results section for Fig 5E (defects in translation) is poorly written, shows modest effects (�1.5-fold), and there is no mechanistic insight that is revealed. 

For example, it is the presise that �RPS26 and �Nob1 strains should both have ribosomes that lack S26. However, their effects on miscoding, stop codon selection, and +1 FS do not phenocopy each other. 

Pno1-KKKF should also generate ribosomes that lack RPS26 – fundamentally therefore, all three strains should have the same effects on translation, and this is clearly not the case. In some cases (start codon selection, miscoding and stop codon readthrough) only KKKF and �RPS26 phenocopy each other; in other cases (+1 FS) �Nob1 and �RPS26 phenocopy each other. These data add little to the story and instead raise a series of questions that are not addressed.

---

## [Decision Letter · Decision Letter 2]

6 Nov 2019

Dear Dr Karbstein,

Thank you for submitting your revised Research Article entitled "A kinase-dependent checkpoint prevents escape of immature ribosomes into the translating pool" for publication in PLOS Biology. We have now obtained advice from two of the original reviewers and have discussed their comments with the Academic Editor who also assessed the revisions in-depth. 

Based on their evaluations, we will probably accept this manuscript for publication, assuming that you will modify the manuscript to address the remaining points raised by the reviewers. Our academic editor emphasises reviewer 3's point 2 and the general point that the polysome westerns and northerns need associated quantification panels. Please also make sure to address the data and other policy-related requests noted at the end of this email.

We expect to receive your revised manuscript within two weeks. Your revisions should address the specific points made by each reviewer. In addition to the remaining revisions and before we will be able to formally accept your manuscript and consider it "in press", we also need to ensure that your article conforms to our guidelines. A member of our team will be in touch shortly with a set of requests. As we can't proceed until these requirements are met, your swift response will help prevent delays to publication.

Sincerely,

Di Jiang, PhD

Associate Editor 

on behalf of 

Hashi Wijayatilake, PhD

Managing Editor

PLOS Biology

DATA POLICY:

Regardless of the method selected, please ensure that you provide the individual numerical values that underlie the summary data displayed in the following figure panels as they are essential for readers to assess your analysis and to reproduce it: Figures 1BC, 2, 3BD, 4CEH, 5D, S4D, S5B, S6B and other panels added in response to reviewer 3's and editorial request described in the main part of this letter. NOTE: the numerical data provided should include all replicates AND the way in which the plotted mean and errors were derived (it should not present only the mean/average values).

Reviewer remarks:

Reviewer #1: Thank you for the opportunity to review this manuscript which establishes how three ribosome assembly factors safeguard cells from immature ribosomes. The authors have addressed all of my previous concerns. This manuscript is acceptable for publication in PLOS Biology.

Reviewer #3: The overall goal of this paper was to establish how three late-stage 40S biogenesis factors (Rio1, Nob1, and Pno1) coordinate to establish a QC checkpoint that prevents the release immature ribosomes into the translating pool. Overall, the authors have revised the manuscript, included new experiments, and addressed most queries that this reviewer had. This manuscript should certainly be published but there remain issues of clarity that could substantially increase the accessibility and dissemination of what has been discovered: 

1. It seems more reasonable to perform the Nob1-D15N OE experiment (Fig 1E) in the Gal::Nob1 (Fig. 1A) background. The absence of 20S rRNA in polysomes could be because of endogenous catalytically active Nob1 in these strains. 

2. Figure 3: The authors have not answered the reviewer’s concern. The reviewer is not concerned about the effect of 20S rRNA in total cells. What is important is that the release of 20S rRNA into polysomes by Rio1-OE in the Nob1-D15N-OE background is modest, and the gels seem to have been deliberately shown with different exposures (empty vector (3C,left), lower exposure; (3C, right) Rio1-OE, longer exposure) to make the phenotype look stronger than the quantitation reveal. To be honest, this remains a general problem in figures showing polysome profiles. As readers we are expected to look at difficult to quantitate westerns and northerns and to evaluate the author’s interpretation in the absence of any quantitation. I understand this is challenging to clarify but it really would be helpful. I really still am struggling in particular with figures 4 and 5 where effects are hard to see and there is no quantitation. Pno1 can be somewhat retained on deep polysomes but this is generally hard to see and not quantitated. Additionally, the model includes no indication that Pno1 is retained on ribosomes that are translating … but I think this is what the data suggest. Perhaps a short discussion in the “results” section of what the results might mean would be helpful. The data are shown, the strict interpretation stated, but I am struggling to fit what I am seeing into a developing model. 

3. For me, the discussion gets very much into the weeds and I am struggling to follow some of the main points, again, mostly connected to the final figures (4/5). It seems that the discussion presents an almost un-weighted summary of all literature, much of it contradictory, and so the strong points of the study get lost. The study makes strong claims and they should be easily deciphered in the final figure and in the summary in the discussion.

4. Figures would be easier to interpret if the relevant information were found attached to each panel (so I don't need to dig into the legend) - for example Fig 4/5 polysomes are all Gal:Rio depleted, but this is not indicated on the actual panel.

---

## [Editor Report · Decision Letter 3]

29 Nov 2019

Dear Dr Karbstein,

On behalf of my colleagues and the Academic Editor, Wendy V Gilbert, I am pleased to inform you that we will be delighted to publish your Research Article in PLOS Biology. 

Early Version

PRESS 

Kind regards,

Hannah Harwood

Publication Assistant, 

PLOS Biology

on behalf of

Hashi Wijayatilake,

Managing Editor

PLOS Biology